# Simultaneous Heavy Metal-Polycyclic Aromatic Hydrocarbon Removal by Native Tunisian Fungal Species

**DOI:** 10.3390/jof9030299

**Published:** 2023-02-24

**Authors:** Neila Hkiri, Dario R. Olicón-Hernández, Clementina Pozo, Chedly Chouchani, Nedra Asses, Elisabet Aranda

**Affiliations:** 1Institute of Water Research, University of Granada, 18071 Granada, Spain; 2Laboratory of Environmental Sciences and Technologies, Higher Institute of Sciences and Technologies of the Environment, University of Carthage, Tunis 1000, Tunisia; 3Laboratorio de Bioquímica y Biotecnología de Hongos, Departamento de Microbiología, Escuela Nacional de Ciencias Biológicas, Instituto Politécnico Nacional, Ciudad de México 07738, Mexico; 4Department of Microbiology, University of Granada, 18071 Granada, Spain; 5Laboratory of Microbial Ecology and Technology, National Institute of Applied Science and Technology, University of Carthage, Tunis 1000, Tunisia

**Keywords:** polycyclic aromatic hydrocarbons, heavy metals, ascomycetes fungi, mycoremediation, phenanthrene, extracellular enzymes, microtoxicity, phytotoxicity, transmission electron microscopy

## Abstract

Multi-contamination by organic pollutants and toxic metals is common in anthropogenic and industrial environments. In this study, the five fungal strains *Chaetomium jodhpurense* (MH667651.1), *Chaetomium maderasense* (MH665977.1), *Paraconiothyrium variabile* (MH667653.1), *Emmia lacerata*, and *Phoma betae* (MH667655.1), previously isolated in Tunisia, were investigated for the simultaneous removal and detoxification of phenanthrene (PHE) and benzo[a]anthracene (BAA), as well as heavy metals (HMs) (Cu, Zn, Pb and Ag) in Kirk’s media. The removal was analysed using HPLC, ultra-high performance liquid chromatography (UHPLC) coupled to a QToF mass spectrometer, transmission electron microscopy, and toxicology was assessed using phytotoxicity (*Lepidium sativum* seeds) and Microtox^®^ (*Allivibrio fisherii*) assays. The PHE and BAA degradation rates, in free HMs cultures, reached 78.8% and 70.7%, respectively. However, the addition of HMs considerably affected the BAA degradation rate. The highest degradation rates were associated with the significant production of manganese-peroxidase, lignin peroxidase, and unspecific peroxygenase. The Zn and Cu removal efficacy was considerably higher with live cells than dead cells. Transmission electron microscopy confirmed the involvement of both bioaccumulation and biosorption processes in fungal HM removal. The environmental toxicological assays proved that simultaneous PAH and HM removal was accompanied by detoxification. The metabolites produced during co-treatment were not toxic for plant tissues, and the acute toxicity was reduced. The obtained results indicate that the tested fungi can be applied in the remediation of sites simultaneously contaminated with PAHs and HMs.

## 1. Introduction

Environmental quality and human health are increasingly being affected by various pollutants of anthropogenic origin. Polycyclic aromatic hydrocarbons (PAHs) and heavy metals (HMs) are some of the most important environmental contaminants (https://www.epa.gov/ accessed on 21 January 2023) and characterised by a high persistence, recalcitrance, abundance, and toxicity [1,2]. The PAHs are mainly released by industrial operations and during the incomplete combustion of organic materials such as coal, fuel, and wood. Exposure to PAHs is associated with various serious diseases due to their teratogenic, carcinogenic, and mutagenic properties. The HMs are widespread in the environment and are mainly derived from industrial wastewater (tanning industry, sewage sludge usage, mining activities, fertilisers, among others). Some HMs, such as Cu (II), Zn (II), and Fe (III), are essential for the biological activities of plants and microorganisms. However, at high concentrations in soils, HMs can be toxic, threatening human and environmental health. The prevalence of typical potentially toxic metals such as Pb, Zn, Cu, Cd, and Ag in paddy soils and in different organs of rice plants in a typical industrial zone in China were investigated [3]. The study demonstrated that the rice containing some heavy metals might cause serious non-carcinogenic and carcinogenic health risks for residents, especially for aged persons. Frequently, PAHs and HMs co-occur, mainly in industrial areas, when different pollutants sources converge in combination with phenomena such as transportation from air and waters from different emission points. Both pollutant types tend to be accumulated in organic matter, making soil a major pollutant reservoir [4].

Among the technologies for the removal of hazardous pollutants such as PAHs and HMs, bioremediation represents an eco-friendly approach in comparison with physical and chemical methods [5,6]. In particular, the use of fungi has several advantages over the use of bacteria since fungi can produce a wide range of unspecific degrading enzymes, possess a large surface-to-cell ratio, and are able to biomineralize or change the valence of HMs [7]. In addition, the interactions between HMs and both live and dead cells lead to a better biological elimination of metals via mechanisms such as cellular surface adsorption, bioleaching, intracellular bioaccumulation, biomineralization, and biotransformation [8]. In the last decades, bioremediation technology studies have focused on the removal of individual PAHs or HMs. For instance, *Aspergillus flavus* and *Aspergillus fumigatus* strains were found to play an important role in bioremediation of 16 PAHs compounds with a degradation rate of 82.7% and 68.9% of the total PAHs after 15 days of incubation [9]. *Podoscypha elegance* strain was also able to degrade 99% of phenanthrene and 98.9% of pyrene in-vitro conditions [10]. In addition, fungal HMs removal is well-documented. For example, the strain *Penicilium simplicissium* showed a great ability to remove Cd, Cr, Cu, Pb, and Zn involving both bioaccumulation and biosorption mechanisms [11]. The removal of Cu, Cr, Cd, and Zn by *Trichoderma brevicompactum* was investigated, which demonstrated a high removal rate for both individual and multi-metal mixture [12].

However, although pollutants rarely occur individually, the bioremediation of sites with various pollutants has rarely been reported [13,14,15]. The coexistence of both pollutant types can negatively affect bioremediation processes since microbes are unable to efficiently bioremediate a co-contaminated environment. As HMs are potent inhibitors of various microbial activities, they can disrupt their regulation and expression by competing for metal binding sites or with enzymes, i.e., oxygenases [16], thus inhibiting the enzyme functions or proteins. Intermediates of PAHs, such as salicylic acid, can also impact microbial cell metabolism, and induce reactive oxygen species (ROS) production, thus affecting the adsorption capacity of HMs [17,18].

The present study investigated the simultaneous PAH biotransformation and HM removal by five selected fungi isolated from polluted Tunisian salt water [19]. Two PAHs, namely phenanthrene (PHE) and benz[a]anthracene (BAA), were used in this work. Four HMs, namely Cu(II), Zn(II), Pb(II), and Ag(II), were tested in mixtures with the PAHs for fungal removal. This study represents a step forward to develop biological processes designed to treat pollutants in complex mixtures.

## 2. Materials and Methods

### 2.1. Chemicals and Strains

The compounds 1-amonibenzothiazole (ABT, 97% purity), 2,2′- azinobis (3-ethylbenzothiazoline-6-sulfonic acid) (ABTS, 98% purity), phenanthrene (PHE, 98% purity), and benz[a]anthracene (BAA, 98% purity) were purchased from Sigma (St. Louis, MO, USA). Copper sulphate and zinc, lead, and silver nitrate metal salts were acquired from Merck (Madrid, Spain). All solvents used were of HPLC grade: acetonitrile (PanReac, AppliChem, Barcelona, Spain), HPLC water (PanReac, AppliChem, Barcelona, Spain), and phosphoric acid (Fisher Chemical, Madrid, Spain). All other chemicals and reagents were of analytical grade or higher purity.

The used fungi were *Chaetomium jodhpurense* (MH667651.1), *Chaetomium maderasense* (MH665977.1), *Paraconiothyrium variabile* (MH667653.1), *Emmia lacerata*, and *Phoma betae* (MH667655.1), previously isolated from a salt environment in Tunisia and molecularly identified [19]. The fungal genera *Chaetomium*, *Paraconiothyrium*, *Phoma,* belonging to *Ascomycota*, were known by their large application in bioremediation due to their great ability to produce extracellular enzymes [20,21,22]. The strain *Emmia lacerate* is a *Basidiomycota* belonging to polypore species. As a white rot fungus it can produce a large wide of extracellular enzymes [23].

### 2.2. Culture Conditions

Mycelium from 5-day-old cultures of the five strains on PDA (potato dextrose agar, BD DIFCO, NJ, USA) petri dish medium was added to 80 mL of sterile distilled water and aseptically homogenised with an ultra-turrax (IKA, Germany) for 10 s. Five hundred μL of the suspension used as pre-inoculum was added into individual flasks containing 25 mL of Kirk’s medium prepared with half-strength artificial seawater [24]. The composition of the artificial seawater was as follows: 29.8 g/L NaCl, 0.73 g/L KCl, 10.7 g/L MgCl_2_•6H_2_O, 5.4 g/L MgSO_4_•7H_2_O, and 1.1 g/L CaCl_2_•2H_2_O. The culture was incubated at 28 °C under agitation at 120 rpm. After 2 days of growth, half of the flaks were heat-inactivated (autoclaved for 20 min at 121 °C) and used as biotic control. All experiments were performed in triplicate.

#### 2.2.1. Phenanthrene and Benz[a]anthracene Biodegradation Experiments

The PHAs (PHE and BAA) were prepared individually in a stock solution (5 mM dissolved in acetonitrile) and added into Kirk’s medium after 2 days of fungal growth to reach final concentrations of 100 µM for PHE and 50 µM for BAA. The experiment was maintained for 18 days at 28 °C under agitation at 120 rpm. Each treatment was performed with three replicates. Flasks were sacrificed at regular intervals of 9 days.

#### 2.2.2. Bioremediation of Combined Pollutants

The influences of HMs on PAH biodegradation and the bioaccumulation properties of the fungi were investigated in a pollutant cocktail of 20 mg/L total concentration, with the addition of a multi-metal mixture [5 mg/L Cu(II), Zn(II), Pb(II), and Ag(II) each] in combination with the PAHs, as previously described in Section 2.2.1. The metal concentration was selected according to a previous experiment to study the minimum inhibitory concentration (MIC) with the selected fungi. A metal stock solution of 1000 mg/L was prepared by dissolving their respective salts CuSO_4_•7H_2_O, ZnNO_3_, Pb(NO_3_)_2_, and Ag(NO_3_) in distilled water, followed by sterilisation by filtration using a syringe filter with a pore size of 0.22 μm. The sterilised metal stock solutions were added separately to the PAH medium flasks to obtain final total metal concentrations of 20 mg/L.

Additionally, to investigate the involvement of CYP450 in the bioremediation processes, the CYP450 inhibitor 1-aminobenzotriazole (ABT) was used at a final concentration of 1 mM in additional flasks, and ABT was added together with the pollutants.

#### 2.2.3. Sampling

After each treatment, half of the content of each flask was centrifuged and stored at −20 °C for enzyme activities, biomass, pH, glucose content, HMs content, Microtox^®^ Bioassay, and phytotoxicity experiments, as described below. Biomass was calculated gravimetrically by filtrating and drying in an oven for 72 h until constant weight. The pH was measured using a reactive strip (Panreac Quimica, Barcelona, Spain).

The remaining content was treated with ethanol (1.6:1 *v*:*v*) to stop the incubation and to obtain the PAHs. After sonication for 15 min, the samples were centrifuged (12,000× *g*), disposed in HPLC glass vials, and analysed by chromatography (HPLC and UHPLC-QTOF).

### 2.3. Enzymatic Activity

Samples of each treatment were filtered and centrifuged at 10,000× *g* for 20 min, and the supernatant was used for enzymatic analyses. Subsequently, Mn peroxidase (MnP), unspecific peroxygenase (UPO), lignin peroxidase (LiP), and laccase (Lac), were analysed on a UV-visible spectrophotometer (Shimadzu UV-1800 UV).

Manganese peroxidase (MnP) activity (EC 1.11.1.13) was monitored at 270 nm through the oxidation of 0.5 mM MnSO_4_ in sodium malonate buffer (50 mM, pH 4.5) with 0.5 mM H_2_O_2_ [25]. Unspecific peroxygenase (UPO) activity (EC 1.11.2.1) was measured using 5 mM veratryl alcohol as substrate in potassium phosphate buffer (50 mM, pH 7.0) in the presence of 1 mM H_2_O_2_ [26]. Lignin peroxidase (LiP) activity (EC 1.11.1.14) was determined by measuring the rate of oxidation of veratryl alcohol toveratryl aldehyde at 310 nm, as described by Tien and Kirk [27]. The mixture reaction contained 2 mM veratryl alcohol and 0.4 mM H_2_O_2_ in sodium tartrate buffer (50 mM, pH 3). Laccase activity (EC 1.10.3.2) was measured using 2,2-azino-bisethylbenthiazolina (ABTS) according to Novotny et al. (1999) [28]. The rate of ABTS oxidation (5 mM) in sodium acetate buffer (0.1 M, pH 4.5), was determined at 420 nm.

One unit of enzyme was defined as the amount of enzyme necessary to release 1 µmol of product per minute under the assay conditions and expressed as UI.

### 2.4. Chemical Analyses

#### 2.4.1. Estimation of Glucose Content

Glucose was analysed according to Miller (1959) [29], using DNS (dinitrosalicyclic acid reagent). The supernatant of each treatment was centrifuged and used in a proportion of 1:1 (sample: DNS reagent). The mixture was boiled at 90 °C for 10 min in a water bath, and after cooling, 1 mL of distilled water was added. The obtained solution was read at a wavelength of 540 nm using a spectrophotometer (Shimadzu UV-1800 UV). The total glucose content was estimated using a standard curve of glucose, and the results are expressed as mg/L.

#### 2.4.2. Phenanthrene and Benzo[a]anthracene Quantification and Metabolite Identification

The removal of PHE and BAA was analysed using an Agilent 1200 HPLC (Agilent, Technologies, Palo Alto, CA, USA), coupled to a DAD detector. The compounds were separated using an RP-C18 Synergy Fusion column (80 Å; 4 μm, 4.6 × 150 mm; Phenomenex^®^, Madrid, Spain). Separation was performed according to the methodology described in Aranda et al. (2009) [30], using water + 1% phosphoric acid (A) and acetonitrile (B) as eluent buffers in isocratic mode, and a flow rate of 1 mL/min. Quantification was performed based on a calibration curve with the pure standards.

The metabolites produced were analysed using an ultra-high performance liquid chromatography (UHPLC), Acquity I-Class System (Waters, Milford, MA, USA), and a Synapt G2S QToF mass spectrometer (Waters, Milford, MA, USA) coupled to a CORTECS-UHPLC^®^ HILIC^®^ C18 1.6 µm column (2.1 × 50 mm; Waters, Milford, MA, USA), connected to a PDA (photodiode array). The flow rate was 0.350 mL/min, using a gradient flow of water, 0.1% NH_3_ (A), and acetonitrile, 0.1% NH3 (B) (5 min A 80%: B 20%; 10 s A 30%: B 70%; 1.40 min A 0%: B 100%; 1.10 min A 80%: B 20%). The data were analysed using the MassLynx software (version 4.1, Waters, Milford, MA, USA). Metabolites were analysed on positive and negative mode.

#### 2.4.3. Heavy Metal Analysis

The removal of Cu, Zn, Pb, and Ag from the culture medium was evaluated by inductively coupled plasma-optical emission spectrometry (ICP-OES) (Perkin-Elmer Optima 8300) at the Center of Scientific Instrumentation of the University of Granada (Granada, Spain). At the end of the incubation time, 10 mL of the samples from HM treatment cultures (biotic control and fungal treatments) were taken and centrifuged at 12,000× *g*. The supernatant was filtered through 0.22-μm Millipore filters and injected into the ICP-OES. Standards with different concentrations were prepared with pure water (Millipore). Recovery value for all metals was 99%.

#### 2.4.4. Transmission Electron Microscopy Coupled with Energy-Dispersive X-ray Spectroscopy (TEM EDX)

Fungal pellets were harvested and washed with phosphate saline buffer (100 mM, pH 6.8). The obtained fungal cells were fixed using 2.5% *v/v* glutaraldehyde overnight at 4 °C, and 2% osmium tetroxide was used as a secondary fixative for 2 h. The fixed cells were washed with 0.1 M phosphate buffer. Subsequently, fungal cells were dehydrated in a series of ethanol, cut into ultra thin sections using an ultramicrotome, and fitted on TEM grids for examination [11]. Samples were visualised using an HR-TEM TALOS F200X (Thermo Fisher Scientific) equipped with a detector of a high-angle annular dark-field (HAADF), Cannon type: Schottky-type field emission (FEG), using the Microanalysis system EDX type (using X-ray energy dispersion) at the Center of Scientific Instrumentation of the University of Granada (Granada, Spain).

### 2.5. Toxicity Bioassays

#### 2.5.1. Phytotoxicity Test

The phytotoxicity test was performed according to the method of Zucconi et al. (1981) [31], using cress seeds (*Lepidium sativum*) to evaluate the toxicity of the culture medium after treatment with the selected fungi. The *L. sativum* seeds were soaked in tap water for 1 h; subsequently, 20 seeds per replicate were positioned at equal distances in glass Petri dishes with Whatman paper Nº 1, and 2 mL of each sample was added at different concentrations (100%, 40%, 20% and 10%), followed by incubation at 28 °C in the dark. Distilled water was used as control, and the abiotic control was prepared using non-inoculated media amended with PHE and BAA, and HMs. After 48 h, the germination index (GI%) was calculated from the number of germinated seeds and the root length, according to the following formula:GI% = (G × L)/G_0_ × L_0_) × 100

G and G_0_: number of germinated seeds in the samples and the control;L and L_0_: root length values in the samples and the control.

#### 2.5.2. Microtox^®^Test

The Microtox^®^ bioassay was employed to evaluate acute toxicity (as EC_50_) of the supernatant after fungal treatment of PAHs and HMs, abiotic, and biotic samples at time 0 and at the end of each experiment. For this, the Microtox^®^ M500 toxicity analyser (Instrumentación Analítica S.A. Madrid, Spain) was employed. Toxicity is expressed as EC_50_ (%), the concentration of the sample that causes a 50% of bioluminescence reduction in the bacterium *Aliivibrio fischeri* after 5 and 15 min of exposure to the sample [32,33]. The analyses were performed in triplicate.

## 3. Results

### 3.1. Removal of Phenanthrene and Benz[a]anthracene

The removal of PHE and BAA was studied in salt Kirk media every 9 days for 18 days. The biodegradation ability of the fungi varied for the different PAHs (Figure 1). For PHE, *E. lacerate* showed the highest biodegradation rate (78.8%), followed by *P. variabile* (74.1%), *C. jodhpurense* (69.3%), *C. maderasense* (48.4%), and *P. betae* (46.3%). However, for BAA, *P. variabile* exhibited the largest removal rate (70.7%) (Figure 1), followed by *P. betae* (58.7%), *E. acerate* (57.8%), *C. maderasense* (28.8%), and *C. jodhpurense* (26.9%).

The influence of the presence of the mixture of HMs (Cu, Zn, Pb, Ag) on PHE and BAA removal was evaluated under the same experimental conditions. The PHE degradation rate varied from 41.1% to 78.0%, and for BAA, the degradation rate ranged between 25.6% and 49.9%.

The degradation rate for BAA coexisting with the mixture of HMs was considerably reduced compared to that in free HM cultures, except for *C. maderasense*, in which the rate of BAA degradation in free HM media increased from 33.7% to 49% in the presence of HMs.

The highest degradation inhibition of BAA in the presence of HMs was observed with *P. betae*, *E. lacerata*, and *P. variabile*, with a degradation rate of approximately 30%. However, the ability of the tested strains to biodegrade PHE coexisting with HMs increased from 48.4% to 51.7% in *C. maderasense*, and from 69.3% to 78.0% in *C. jodhpurense*. A moderate decline in the PHE biodegradation rate by approximately 3% was observed in *E. lacerata*, and 5% in *P. betae* and *P. variabile* was detected in the presence of HMs.

The role of the cytochrome P450 enzymatic system (CYP) in the degradation of the co-contaminated media was evaluated by the addition of the inhibitor 1-aminobenzotriazole. A lower degradation activity was observed in cultures containing CYP inhibitor with only the *P. betae* strain, in which the degradation rate decreased by 12.8% and 14.6% for PHE and BAA, respectively. This indicates that *P. betae* is involved in both extra- and intracellular enzymatic PAH removal.

For the other strains (*C. jodhpurense*, *C. maderasense*, *P. variabile* and *E. lacerata*), inhibition of the CYP did not significantly influence the biodegradation rate of PAHs (Figure 1), indicating that PHE and BAA could be transformed mainly via extracellular pathways.

Regarding the pH profile, glucose use, and biomass production by fungi in the absence and presence of HMs in the growth medium, a shift of pH from 5 to 7 and an increase in biomass accumulation during the first 9 days with or without HMs were detected. A rapid decrease in the glucose concentration was also observed in the first 9 days, followed by a moderate and stable decline in the remaining 9 days of incubation. The biomass produced by the fungi was increased considerably. The presence of HMs did not affect fungal growth throughout the experiment. However, in the last 9 days, biomass accumulation in *P. betae* and *C. jodhpurense* increased from 8 g/L in the PAH treatment to 10 g/L in the presence of HMs.

### 3.2. Enzyme Production during Degradation

Extracellular ligninolytic enzyme production was also monitored after 9 and 18 days of incubation in Kirk media supplemented with the PAHs, and in the presence or absence of HMs (Table 1).

The production of MnP, LiP, UPO, and laccase enzymes was detected in most of the treatment cultures of the five tested fungi (Table 1).

Maximum MnP production was observed in *C. maderasense*, which reached 274.0 ± 12.3 UI in the PAH treatment. The presence of HMs with PAHs negatively affected enzyme activity; MnP was not detected in *C. jodhpurense*, and its level decreased considerably in *C. maderasense* (from 274.0 ± 12.3 UI to 120.6 ± 5.7 UI). The highest level of production was found after 9 days of incubation.

Among the peroxidase enzymes, UPO had the highest levels. The fungus *P. variabile* produced the maximum amounts of 329.4 ± 15.9 UI in the presence of HMs, and of 309.4 ± 41.2 UI in PAH media. Co-exposure to HMs and PAHs enhanced UPO activity three-fold in *C. jodhpurense* and four-fold in *E. lacerate*; LiP activity was lower than MnP and UPOs. The highest value was detected in *P. variabile* (200.4 ± 37.6 UI). In treatments amended with HMs, LiP activity increased in *C. maderasense* (from 15.2 ± 1.4 UI to 175.7 ± 36.2 UI), *P. variabile* (from 55.4 ± 5.6 UI to 200.4 ± 37.6 UI) and *E. lacerata* (from 41.8 ± 12.1 UI to 71.4 ± 13.8 UI). In *C. jodhpurense* and *P. betae*, LiP activity decreased from 41.55 ± 6.4 UI to 12.2 ± 2.8 UI and from 65.0 ± 11.4 UI to 45.5 ± 10.2 UI, respectively.

Only *C. maderasense* and *P. betae* produced considerable levels of Lac in both PAH treatment and PAHs with HMs. Interestingly, the addition of HMs to PAHs in *P. betae* increased Lac production 14- and 8-fold, respectively, after 9 and 18 days of cultivation.

### 3.3. Metabolic Products after PHE and BAA Biodegradation

The intermediate metabolites of PHE and BAA after biodegradation by the strains detected by LC-MS are shown in Figure 2. No significant differences were observed in the metabolism of the PAHs among the strains or between the presence of ABT and HMs. The analysis revealed the presence of 1-phenanthrol (P1), phenanthrene 9,10-dihydrodiol (P2), anthracene (B1), anthrone (B2), anthraquinone (B3), phthalic anhydride (B4), and phthalic acid (B5), being the main metabolites produced under the tested conditions, although we also found other unspecific metabolites (Appendix A Table A1). Phthalic anhydride and phthalic acid could be part of both BAA and PHE degradation metabolic pathways as a consequence of ring cleavage reactions under in vivo conditions after the first oxidation steps. The hydroxylated metabolites confirm the action of peroxidases in PAH biotransformation.

### 3.4. Heavy Metal Removal

#### 3.4.1. Residual Heavy Metal Content

Figure 3 shows the residual metal (Cu, Zn, Pb and Ag) concentrations in the culture media. After 18 days of cultivation, both live and inactivated mycelia showed high removal efficiencies for Ag and Pb. The highest amounts of Ag(II) and Pb(II) absorbed by inactivate mycelium were 4 and 4.4 mg/L, respectively. These amounts were increased to reach 4.9 mg/L with live cells at the end of the experiment. However, only live cells removed large amounts of Zn(II) and Cu(II), with a maximum of 4.7 mg/L for Zn(II) and 3.9 mg/L for Cu(II), contrary to inactivated mycelia, which did not exceed 0.357 mg/L for Cu and 0.801 mg/L for Zn. At the end of the experiment, after 18 days of incubation, the total amounts of metals removal by live cells were 4.9 mg/L for Ag(II) and Pb(II), 4.7 mg/L for Zn(II), and 3.9 mg/L for Cu(II).

#### 3.4.2. Transmission Electron Microscopy Coupled with EDX Analysis

Figure 4 shows the transmission electron micrographs of the five fungi. The TEM images clearly show the morphological structures of the fungi after exposure to HMs. The EDX spectrum of strain *C. maderasense* shows only the presence of Cu and Zn inside the cell, reflected by the presence of some dark areas (Figure 4a). Ultrastructure analysis of *C. jodhpurense*, with the presence of an intact and regular cell membrane and a well-organised cytoplasm distribution (Figure 4b), and EDX analysis revealed no interesting findings in relation to the tested metals. The dark electron granules (Figure 4c) found in *P. variable* in the cell wall/cell membrane and in the cytoplasm were identified by EDX as Cu. Dark, and dense spots were observed in various areas in the cytoplasm of *E. lacerate*, and EDX analysis confirmed the prevalence of the four metals (Cu, Zn, Pb, and Ag) (Figure 4d), indicating bioaccumulation. Strain *P. betae* accumulated Cu, Zn, and Ag throughout the cell wall/cell membrane, and the cytoplasm showed visible dark spots inside the fungal cell. The cell wall/cell membrane was surrounded by electron-dense granules identified by the EDX spectrum as Pb (Figure 4e). The obtained results indicate that this tolerance towards Cu, Zn, Pb, and Ag could be associated with both biosorption and bioaccumulation mechanisms.

### 3.5. Ecotoxicological Studies

#### 3.5.1. Phytotoxicity Analysis

The plant species *L. sativum* was used as indicator plant to test the toxicity of the bioremediated culture medium. Seeds were assigned to an abiotic control (untreated mixture of PAHs and HMs), inoculated with PAHs in the presence and absence of HMs with dead and live cells of the selected fungi, without dilution and with a serial of dilution (0.4, 0.2, and 0.1) (Figure 5). Media containing untreated culture medium negatively affected the growth of *L. sativum*, with a germination index of 1.7% ± 0.8% and 49.1% ± 1.1%, in 0.4 and 0.2, respectively; at higher concentrations, germination was inhibited completely. Untreated systems were toxic and inhibited the germination of *L. sativum*.

Samples obtained after PAH treatment in the presence or absence of HMs with either fungal species revealed a decrease in phytotoxicity, depending on the dilution.

The highest germination index observed in PAH culture medium with a dilution of 0.2 was 99.5% ± 9.3% with *P. betae*, followed by *C. judheprenses* (87.5% ± 3.3%), *P. variabile* (81.9% ± 2.3%), *E. lacerata* (73.2% ± 3.3%), and *C. maderasense* (72.7 ± 7.2%). In cultures amended with a mixture of HMs and PAHs, the highest phytotoxicity reduction was observed for the 0.2 dilution, and the highest germination index was 96.5% ± 7.8% with *C. maderasense*, followed by *P. betae* (88.3% ± 7.8%); for the other strains, the germination index was approximately 66%.

Treatment with inactivated mycelia was less effective. The germination index did not exceed 66.5% with all tested fungi, indicating a prevalence of biological transformation rather than bioadsorption. This leads us to infer that live mycelia not only removed PAHs and HMs, but also potentially transformed toxic compounds into less hazardous products.

#### 3.5.2. Microtox^®^Test

The Microtox^®^ bioassay represents an additional tool to assess the acute toxicity of pollutants and mixtures, but also of their degradation metabolites. High toxicity was observed with untreated culture medium (AC) (EC_50_ 5 min = 5.5% ± 0.4%; EC_50_ 15 min = 4.6% ± 0.1%) at the beginning of the experiments and after 18 days (EC_50_ 5 and 15 min = 2.7% ± 0.1%) (Figure 6). The toxicity levels of cultures with HMs and PAHs in mixture were almost similar for all samples at the onset of the experiments, except for culture media with live cells (MXT) of *P. variabile* (Figure 6c). Only the treatment with dead cells (IM) of E. lacerate, incubated over 18 days, exhibited a slight reduction in acute toxicity (Figure 6d).

However, with live cells (MXT), a robust EC_50_ decrease was found for cultures treated with *E. lacerata* (Figure 6d) and *P. betae* (Figure 6e) (up to EC_50_ = 32%). Treated cultures with *C. maderasense* (Figure 6b) and *P. variabile* (Figure 6c) also exhibited a decrease in acute toxicity after 18 days of incubation (up than EC_50_ = 23%). On the other hand, *C. jodhpurense* was the only fungal strain that did not show toxicity reduction, irrespective of the exposure time (EC_50_ 5 min = 5.6% ± 0.1%; EC50 15 min = 5.7% ± 0.1%, respectively) (Figure 6a).

## 4. Discussion

The metabolism of fungi in response to complex pollution stress has received little attention [34]. Usually, PAHs and HMs are found alongside in contaminated environments such as manufacturing plants and refinery sites, and their presence may affect biodegradation processes by impacting both the physiology and ecology of degrading microorganisms, and by competing with metal cofactors and enzymes, consequently reducing the enzymatic activity of degrading microorganisms [35]. Xu et al. (2022) [36] discussed the impact of organic carbon on the redox transformation of toxic metals. Due to their redox moieties such as phenol or quinine, organic carbon can affect redox reaction of some HMs such as As, Cr, and Hg. On the other side, HMs can affect enzymatic PAHs degradation. Excess HMs can compete with macronutrients serving as cofactors for enzymes, thus inhibiting the enzymes activities. They can also denature enzymes by combining with their sulfhydryl groups [37].

The tested fungi were grown during 2 days before the addition of contaminants to support growth. The degradation of PHE and BAA was studied in different systems: in the presence or absence of HMs, and with the inhibition of the intracellular enzymatic system. The obtained results showed that fungal strains could tolerate and degrade PAHs, as evidenced by the high biomass production and glucose consumption. Notably, the addition of glucose, which served as a carbon energy source, facilitated fungal growth and PAH degradation, indicating the co-metabolic trait of all strains.

The five tested fungi showed high biodegradation abilities, ranging from 46.3% to 78.8% for PHE, and from 26.9% to 70.7% for BAA. The addition of ABT, a CYP inhibitor, decreased the degradation of PHE and BAA only in *P. betae*, suggesting that CYPs play an important role in PAH degradation in this strain. Li et al. (2018) [38] found that the inhibition of CYPs suppressed the degradation of PHE but not that of BAA.

The biodegradation efficiency for BAA decreased slowly In the presence of HMs in *P. betae*, *P. variabile*, and *E. lacerate* from 58.1% to 25.6%, 57.4% to 29.1%, and 70.7% to 38.4%, respectively. This might have been caused primarily by the addition of the tested HMs since they could stress metabolic activities and interact with various key fungal activities such as ATP production, substrate mineralization, and cell surface functions, which play important roles in the transport, biotransformation, and detoxification of organic xenobiotics [37].

However, all tested strains tended to degrade PHE with approximately the same rates as those observed for HM-free cultures. Hong et al. (2010) [39] investigated *Fusarium solani* and *Hypocera lixii*, isolated from petrol station soil, for pyrene degradation as well as Cu and Zn tolerance. Both isolated strains degraded more than 60% of pyrene, with the accumulation of Cu and Zn. Janicki et al. (2018) [6] observed that the presence of Pb and Zn individually decreased xenobiotic removal rates.

In the present work, the pH increased from 5.5 to 7.5–8, which is in agreement with the findings of Ye et al. (2011) [40], who demonstrated that anthracene degradation by *A. fumigatus* was optimal within a pH range of 5–7.5.

The biodegradation process was associated with the production of different extracellular enzymes, mainly peroxidases and laccases, which were produced concomitantly to the reduction inthe pollutants. The highest degradation rates were found in strains producing significant amounts of MnP, LiP, and UPOs. Only *P. beate* and *C. maderasense* produced laccase, and these strains were not among those with the highest degradation rates. These results suggest that peroxidases play a main role in PHE and BAA degradation by the tested fungi. Ye et al. (2011) [40] stated that LiP is the principal enzyme involved in the degradation of anthracene. However, according to Li et al. (2018) [38], laccase plays the main role in PHE and BAA transformation in the white rot fungus *Pycnoporus sanguineus*. Many enzymes are regulated by HMs at transcription and activity level, and therefore, in most cases, metal ions might inhibit the activities of enzymes involved in the degradation of the tested compounds [41]. Moreover, our findings indicate that metal supplementation increased laccase production in *P. betae*. This could be explained by the presence of Cu, which is a well-known strong laccase inducer [42].

The same metabolites were found in different testing conditions and with the five strains, and the following conclusions were derived: first, the presence of HMs did not affect enzymatic activities, and second, the ligninolytic enzymes and CYPs could transform PHE and BAA to the same products, which is in agreement with previous findings by Luo et al. (2016) [43] and Li et al. (2018) [38].

In the present study, the degradation of PHE and BAA generated phthalic acid and benzoic, both of which are metabolites of the degradation of different PAH molecules by ligninolytic enzymes [44,45,46].

Our results clearly show that the tested fungal strains can tolerate a mixture of HMs and remove them successfully via adsorption (dead cells) and trough metabolic processes (live cells). The removal efficacy was better with live cells, indicating the presence of active, energy-dependent transporters [8,47]. After adsorption onto the cell wall, and unlike for dead cells, live cells can bioaccumulate metal ions on their cell surface for sequestration and detoxification [48]. Dead cells showed a high uptake efficacy for Pb, followed by Ag. On the contrary, only live cells removed large amounts of Cu and Zn. The fungal uptake preference in a multi-metal medium depends on the metal properties such as electronegativity, ionic radius, and hydration energy [49,50]. The metals Pb and Ag possess a higher electronegativity (2.33 and 1.93 Pauling, respectively) and a larger ionic radius (1.75Åand 1.44Å, respectively) than Cu (1.90 Pauling; 1.28 Å) and Zn (1.38 Pauling; 1.65 Å). This could explain the great affinity of dead cells for Pb and Ag. Several studies described Pb(II) as the metal most easily removable by filamentous fungi [11,51,52]. Adsorption processes are related to the composition of the mycelium cell wall without energy involvement, whereas extracellular polymeric substances play an important role in heavy metal adsorption at binding sites through different mechanisms such as proton exchange [53].

The bioremediation of PAHs and HMs can be affected by the nature of the contaminants or the interaction of these two pollutant types [37]. The PAHs can change the permeability of biomembranes by interacting with the lipophilic components of the fungal cytoplasm, which promotes the entry of HMs into the cells, and may affect cellular functions [34]. Dey et al. (2020) [35] found that, in the presence of a pesticide, *A. fumigatus* showed a higher Pb and Zn removal capacity.

The TEM-EDX micrographs confirmed that fungal strains exhibited various responses against HM stress, suggesting the involvement of both bioaccumulation and biosorption. The heterogeneous responses of the different tested fungi might be due to their various resistance mechanisms and tolerance strategies against HMs, such as efflux system, extracellular precipitation, chemical transformation, mycelial cell wall compositions, cell surface adsorption, and intracellular accumulation [8,54,55]. Intact cell structures observed in the TEM photomicrographs, with regular cytoplasm dispersion and a well-defined cell membrane, confirmed the high abilities of the tested fungi to resist and survive under co-contaminant stress. This is supported by the findings of Dey et al. (2020) [35], who underlined the ability of *A. fumigatus* for HM removal from a multi-metal pesticide matrix.

Investigations on the degradation and elimination of metals and organic chemicals by microorganisms should be coupled with detoxification assays. In this work, the phytotoxicity test confirmed that all tested fungi produced non-toxic metabolites during co-treatment of PAHs and HMs at a dilution of 0.2. Along with this, the Microtox^®^ bioassay revealed acute toxicity decreases (especially in the treatments with live cells) with all tested strains except for *C. jodhpurense*. This finding correlates with the lowest BAA removal rate (22.91%) observed for the same strain. In no case did the EC_50_ values recorded after fungal treatment exceed 33%, most likely because of the high toxicity of PAHs and HMs in untreated culture media at the beginning of the experiments (EC_50_ 5 min = 5.5% and EC_50_ 15 min = 4.6%).

## 5. Conclusions

This study investigated the ability of different fungal strains to bioremediate environments co-contaminated with PAHs (PHE and BAA) and heavy metals. The strains could efficiently degrade phenanthrene and benzo[a]anthrancene, along with four heavy metals (Cu, Zn, Pb, and Ag). Bioremediation is linked with ligninolytic enzyme production, which may lead to the generation of phthalic and benzoic acid as PHE and BAA metabolites. The toxicological analyses confirmed the detoxification of emerging pollutants. Overall, the tested fungal strains are suitable candidates for the environmentally friendly remediation of co-contaminated sites.

## Figures and Tables

**Figure 1 jof-09-00299-f001:**
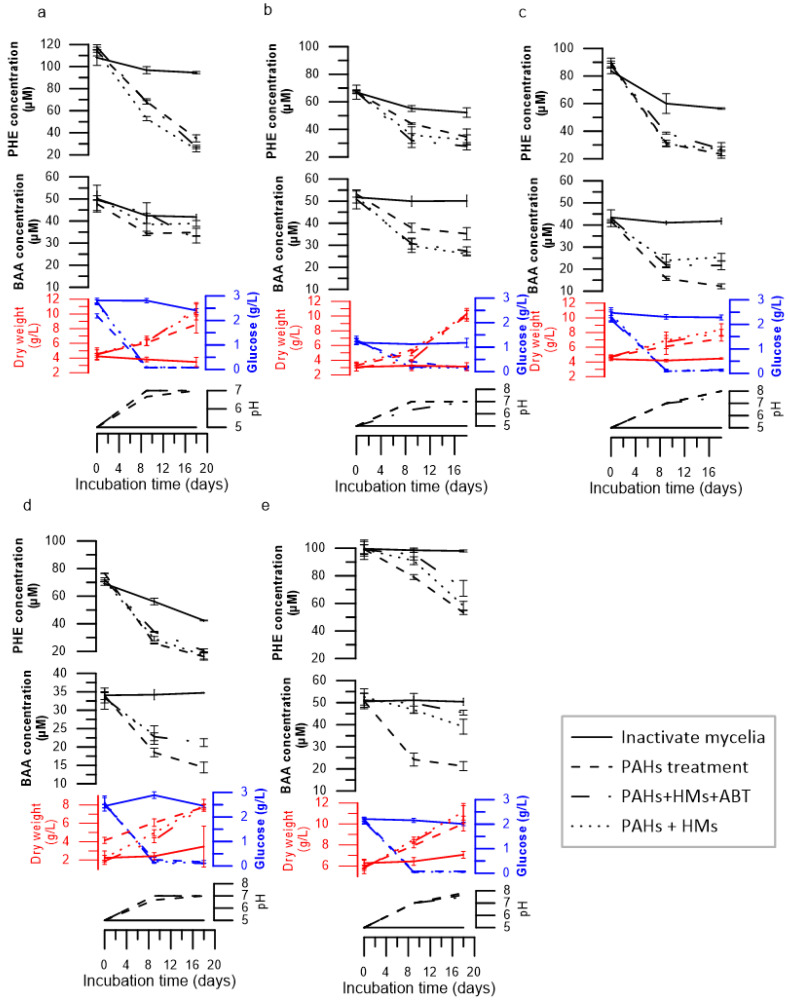
Residual phenanthrene and benzo[a]anthracene (μM) in the culture medium, fungal biomass (g/L) (red color), pH variation, and glucose concentration (g/L) (blue color). (**a**) *C. jodhpurense*, (**b**) *C. maderasense*, (**c**) *P. variabile*, (**d**) *E. lacerata*, and (**e**) *P. betae*. Error bars indicate the average of tree replicates (n = 3).

**Figure 2 jof-09-00299-f002:**
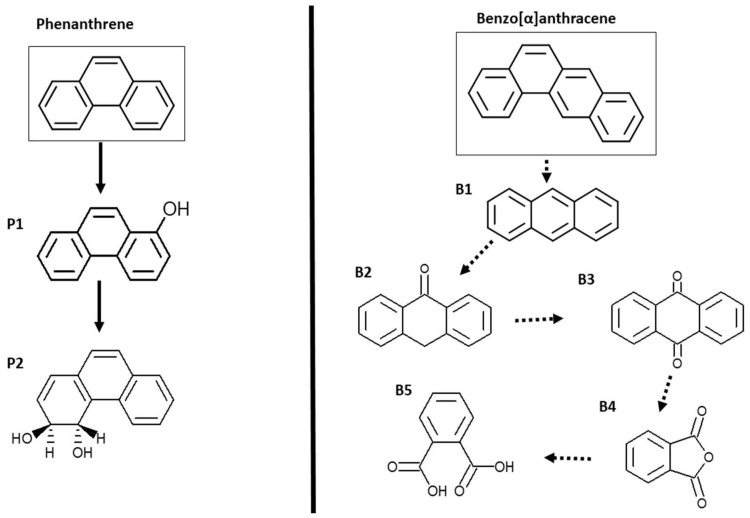
Proposed pathways for the biotransformation of phenanthrene and benz[a]anthracene by the selected fungi. The same metabolites were found with the five tested strains for all experimental variants.

**Figure 3 jof-09-00299-f003:**
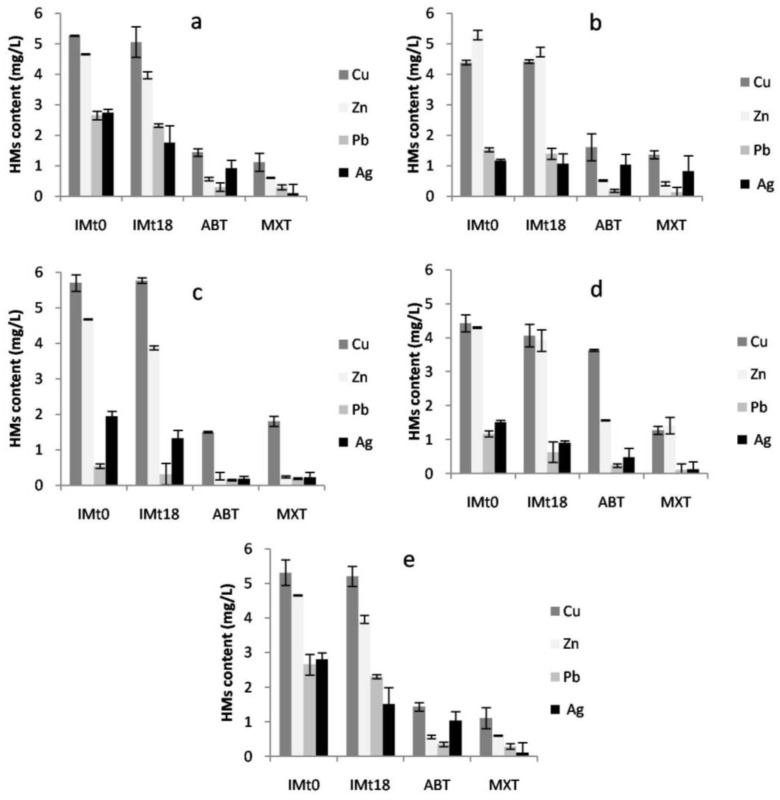
Residual concentrations of Cu, Zn, Pb, and Ag, expressed in mg/L in the culture medium of inactivate mycelium (IM), and for each fungus in the presence of PAH (MXT) and ABT for (**a**) *C. jodhpurense*, (**b**) *C. maderasense*, (**c**) *P. variabile*, (**d**) *E. lacerata*, and (**e**) *P. betae* after18 days of incubation. Error bars indicate the average of tree replicates (n = 3).

**Figure 4 jof-09-00299-f004:**
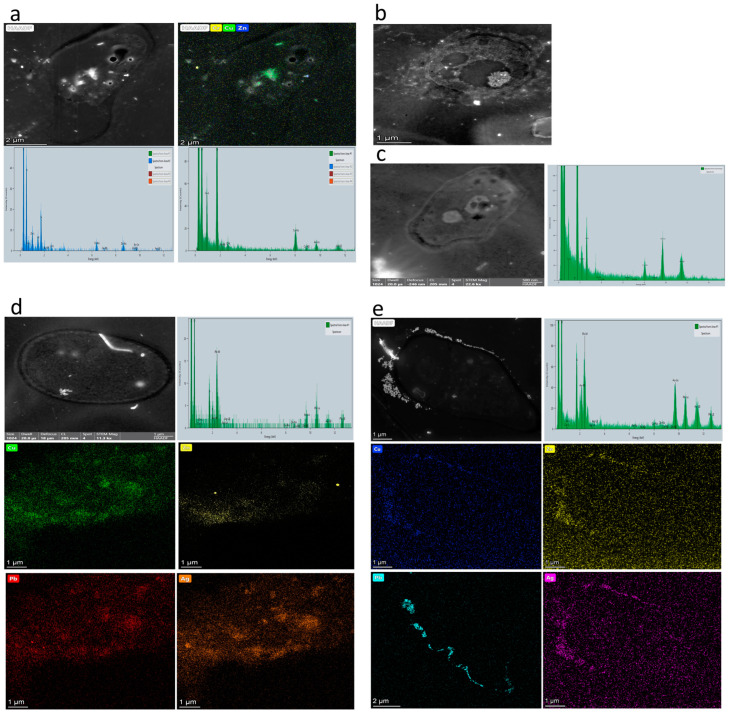
TEM of the five fungal strains in the presence of a multi-metal cocktail after 9 days of incubation. (**a**) *C. maderasense*, (**b**) *C. jodhpurense*, (**c**) *P. variabile*, (**d**) *E. lacerata*, and (**e**) *P. betae* in the presence of 20 mg/L of each Cu, Zn, Pb, and Ag.

**Figure 5 jof-09-00299-f005:**
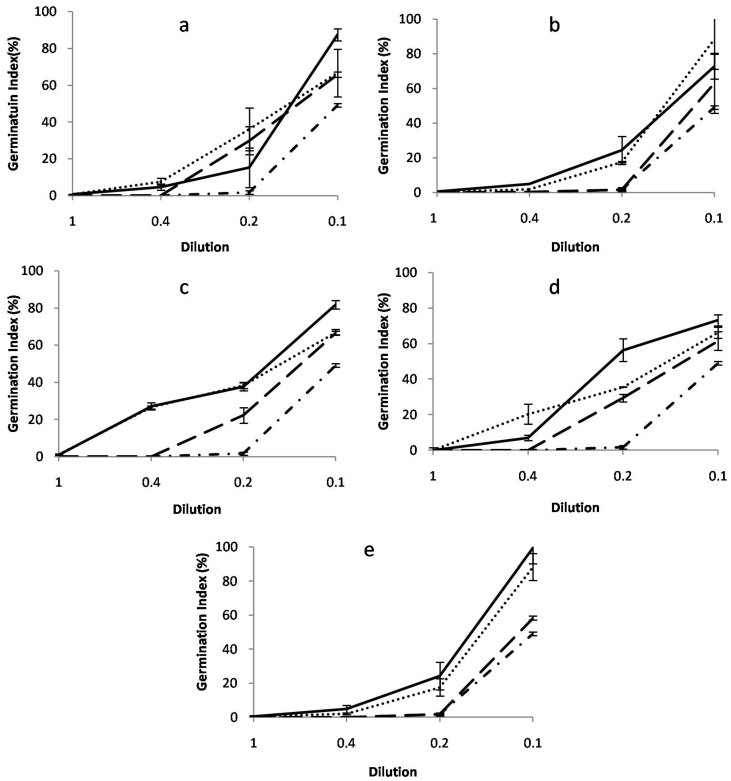
Phytotoxicity effects of PAHs and HMs on *L. sativum*, expressed as the germination index (GI) at 48 h after treatment with PAHs (solid line), PAHs and HMs (dotted line), inactivated mycelium (long-dash line), and untreated PAHs and HMs (dotted dash line), with (**a**) *C. jodhpurense*, (**b**) *C. maderasense*, (**c**) *P. variabile*, (**d**) *E. lacerata*, and (**e**) *P. betae*. Error bars indicate the average of tree replicates (n = 3).

**Figure 6 jof-09-00299-f006:**
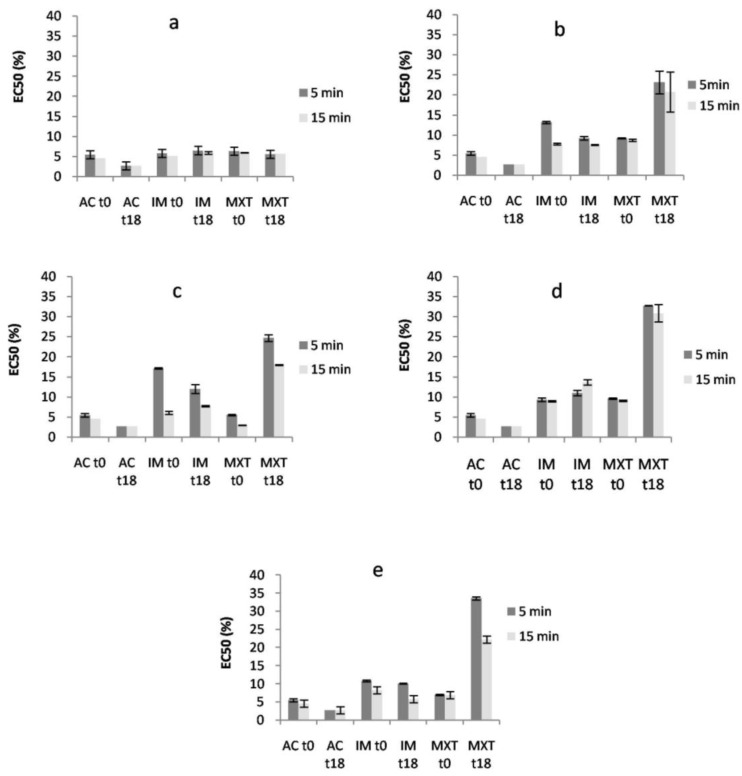
Acute toxicity (as EC_50_ after 5 and 15 min after exposition) for (AC) untreated media, (IM), PAH and HM cultures treated with inactivate mycelia, and (MXT) treatment of PAH and HM culture media with live cells of (**a**) *C. jodhpurense*, (**b**) *C. maderasense*, (**c**) *P. variabile*, (**d**) *E. lacerata*, and (**e**) *P. betae* at time 0 and after 18 days of incubation. Error bars indicate the average of three replicates (n = 3).

**Table 1 jof-09-00299-t001:** Enzymatic activities. MnP—manganese peroxidase; UPO—unspecific peroxygenase; LiP—lignin peroxidase; Lac—Laccase, expressed as (UI) during PAH treatment after 9 and 18 days of cultivation in the absence and presence of HMs. ±Value indicates the average of tree replicates (n = 3).

		MnP		UPO		LiP		Lac	
		T9	T18	T9	T18	T9	T18	T9	T18
** *C. jodhpurense* **	PAH	87.6 ± 7.4	16.0 ± 1.4	n.d.	41.5 ± 6.4	32.4 ± 10.3	25.7 ± 1.8	n.d.	8.8 ± 2.5
	PAH + HM + ABT	8 ± 2	n.d.	n.d.	8.98 ± 1.94	16.6 ± 3.9	43.2 ± 9.2	n.d.	8.4 ± 0.9
	PAH + HM	n.d.	n.d.	n.d.	12.2 ± 2.8	23.9 ± 3.8	99.4 ± 3.0	n.d.	8.0±1.3
** *C. maderasense* **	PAH	274.0 ± 12.3	9.1 ± 4.5	15.2 ± 1.4	17.8 ± 2.3	n.d.	n.d.	43.7 ± 5.1	225.1 ± 13.8
	PAH + HM + ABT	259 ± 12	9.0 ± 1.7	26.0 ±0.5	n.d.	10.2 ± 2.3	n.d.	n.d.	n.d.
	PAH + HM	120.6 ± 5.7	3.6 ± 0.7	175.7 ± 36.2	n.d.	10.9 ± 2.4	n.d.	n.d.	122.1 ± 3.5
	PAH	14.9 ± 0.9	85 ± 14	55.5 ± 5.6	37.9 ± 6.1	309.1 ± 41.2	n.d.	n.d.	n.d.
** *P. variabile* **	PAH + HM + ABT	88.7 ± 3.9	66.1 ± 6.2	77.2 ± 4.8	n.d.	111 ± 31	n.d.	n.d.	n.d.
	PAH + HM	51 ± 11	7.4 ± 2.8	200.4 ± 37.6	181.4 ± 17.0	329.4 ± 15.9	n.d.	n.d.	n.d.
** *E. lacerata* **	PAH	25.0 ± 3.7	n.d.	n.d.	41.8 ± 12.1	62.2 ± 4.6	42.6 ± 2.7	29.3 ± 1.7	n.d.
	PAH + HM + ABT	n.d.	66.5 ± 26.1	n.d.	105.7 ± 7.3	66.2 ± 5.7	46.6 ± 5.5	n.d.	n.d.
	PAH + HM	n.d.	15.6 ± 2.3	n.d.	71.4 ± 13.8	183.9 ± 25.4	62.9 ± 6.7	n.d.	0
** *P. betae* **	PAH	4.2 ± 0.9	25.4±8.4	21.6 ± 2.4	65.0 ± 11.4	19.5 ± 2.0	69.4 ± 4.7	47.3 ± 3.9	63.5 ± 6.5
	PAH + HM + ABT	24.7 ± 3.2	25.9 ± 3.4	22.4 ± 2.4	10.5 ± 1.3	9.1 ± 2.3	77.4 ± 10.5	65.6 ± 11.8	39.1 ± 0.9
	PAH + HM	18.5 ± 2.7	n.d.	27.3 ± 3.5	45.5 ± 10.2	32.8 ± 6.9	76.0 ± 11.9	687.5 ± 28.3	523.4 ± 21.5

n.d. = not detected.

## Data Availability

Not applicable.

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
