# Peer review of "Simultaneous Heavy Metal-Polycyclic Aromatic Hydrocarbon Removal by Native Tunisian Fungal Species"

_jof, 2023, doi:10.3390/jof9030299_

Round 1

Reviewer 1 Report

The manuscript of jof-2201226 entitled “Simultaneous heavy metal-polycyclic aromatic hydrocarbon removal by native Tunisian fungal species” made an important contribution to understanding the ability of different fungal strains to bioremediate environments co-contaminated with PAHs (PHE and BAA) and heavy metals. The strains could efficiently degrade phenanthrene and benzo[a]anthrancene along with four heavy metals (Cu, Zn, Pb and Ag). Bioremediation is linked with ligninolytic enzyme production, which may lead to the generation of phthalic and benzoic acid as PHE and BAA metabolites. The toxicological analyses confirmed the detoxification of emerging pollutants. Overall, the tested fungal strains are suitable candidates for the environmentally friendly remediation of co-contaminated sites.The study falls well into the scope of jof. However, before the manuscript could be accepted by the journal, some concerned issues should be addressed properly.

Detailed minor comments:

Abstract:

It lacks of the media information for five fungal strains HM removal and PAH degradation, it is solution or other media, which needs to be specified.

The illustration of "The heavy metals Pb and Ag were efficiently removed by both live and death cells, whereas Zn and Cu were only removed by live cells." benefit from providing specific experimental data.

Introduction:

Lines 50-52, the description is not sufficient to cover the target elements of Cu(II), Zn(II), Pb(II) and Ag(II), especially the Ag(II), which is very unique.In addition, the recent related publication of "Sci. Total Environ.2023,856,158883." is suggested to be included to make the illustration comprehensive and up to date.

Materials and Methods:

Lines 100, readers may feel confused about "with 50% of artificial sea water", which needs clarifying.

Lines 102,  CaCl2, format problem.

Lines 114-115, based on the individual concentration of 4 elements of 5 mg/L each, 20 mg/L total concentration was obtained? These description seems inconsistent with lines of 118-122. Please check.

Line 185, the unit for 12,000 is not common.

2.4.3. Heavy metal analysis, it is suggested to provide data control measures including elemental recoveries, etc.

Results:

Lines 225-226,"The removal of PHE and BAA was studied in salt Kirk media every 9 days for 18 days." why only two time scale? The tendency after 18 days seems unclear. It needs clarification.

Line 376, "n.d.= non detected." should be traditionally "not detected".

Discussion:

Lines 427-431, The inclusion of the recent related publication of "Carbon Res. 1, 9. (https://doi.org/10.1007/s44246-022-00010-8)" may help consolidate the discussion.

Author Response

Dear Reviewer,

We appreciate your comments and corrections to improve the manuscript. Please find attached to this letter the files of the revised version of the abovementioned manuscript, to be considered for publication. We checked the whole manuscript again for typing errors to improve the manuscript. Changes in the manuscript have been marked in yellow.

Detailed comments to each of the reviewers’ remarks are given below. The original reviewers´ remarks are written in regular font while our answers/ statements are written in bold. The paper has been revised by proofreading services.com.

Reviewers 1

The manuscript of jof-2201226 entitled “Simultaneous heavy metal-polycyclic aromatic hydrocarbon removal by native Tunisian fungal species” made an important contribution to understanding the ability of different fungal strains to bioremediate environments co-contaminated with PAHs (PHE and BAA) and heavy metals. The strains could efficiently degrade phenanthrene and benzo[a]anthrancene along with four heavy metals (Cu, Zn, Pb and Ag). Bioremediation is linked with ligninolytic enzyme production, which may lead to the generation of phthalic and benzoic acid as PHE and BAA metabolites. The toxicological analyses confirmed the detoxification of emerging pollutants. Overall, the tested fungal strains are suitable candidates for the environmentally friendly remediation of co-contaminated sites.The study falls well into the scope of jof. However, before the manuscript could be accepted by the journal, some concerned issues should be addressed properly.

Detailed minor comments:

 Abstract:

1) It lacks of the media information for five fungal strains HM removal and PAH degradation, it is solution or other media, which needs to be specified.

We added the media as recommended

2) The illustration of "The heavy metals Pb and Ag were efficiently removed by both live and death cells, whereas Zn and Cu were only removed by live cells." benefit from providing specific experimental data.

The illustration was rectified it become “Zn and Cu removal efficacy was considerably higher with live cells than dead cells”.

 Introduction:

3) Lines 50-52, the description is not sufficient to cover the target elements of Cu(II), Zn(II), Pb(II) and Ag(II), especially the Ag(II), which is very unique.In addition, the recent related publication of "Sci. Total Environ.2023,856,158883." is suggested to be included to make the illustration comprehensive and up to date.

We included a paragraph as recommended in Lines 54-58,

Materials and Methods:

4) Lines 100, readers may feel confused about "with 50% of artificial sea water", which needs clarifying.

Line 100 will be line 120 after modification. We replaced” with 50% of artificial sea water” by “with Half-strength artificial seawater”

5) Lines 102,  CaCl2, format problem.

Corrected accordingly (Line 121)

6) Lines 114-115, based on the individual concentration of 4 elements of 5 mg/L each, 20 mg/L total concentration was obtained? These description seems inconsistent with lines of 118-122. Please check.

It was rectified “1000 mg/L instead of 1,000 mg/L” (L 139)

The word total was added to the expression “obtain final metal concentrations of 20 mg/L” which become “obtain final total metal concentrations of 20 mg/L”. (L141)

7) Line 185, the unit for 12,000 is not common

We apologize, we meant g-force, that is often used instead of rpm because the size of the rotors will be different depending on the brand and therefore the g-force will be different. We have removed de x to avoid confusion.

8) 2.4.3. Heavy metal analysis, it is suggested to provide data control measures including elemental recoveries, etc.

A sentence was added: “Recovery value for all metals was 99%”

Results:

9) Lines 225-226,"The removal of PHE and BAA was studied in salt Kirk media every 9 days for 18 days." why only two time scale? The tendency after 18 days seems unclear. It needs clarification.

Based on previous experiments the best result was obtained at 18 days. After 18 days the performance of tested fungi decreased and degradation rate become stable.

Three time scale (time 0, time 9 and time 18) were chosen because we don’t need a kinetic of degradation and since we was using sacrificed flaks, with five fungi in four different condition cultures in triplicate we got 60 samples for each measuring point, so we considered 3 data points relevant for our study.

10) Line 376, "n.d.= non detected." should be traditionally "not detected".

Corrected accordingly (Line 405)

 Discussion:

11) Lines 427-431, The inclusion of the recent related publication of "Carbon Res. 1, 9. (https://doi.org/10.1007/s44246-022-00010-8)" may help consolidate the discussion.

Discussion was consolidate as recommended (Lines 452-458)

Reviewer 2 Report

jof-2201226-peer-review-v1

Simultaneous heavy metal-polycyclic aromatic hydrocarbon removal by native Tunisian fungal species.

Some suggestions are mentioned below

1-    Abstract

Please include the strain number of each of them

2-    2.1. Chemicals and strains

Please include more information about the strains,

Define once what MH means

3-    Lines 21-22, in pag1

……were investigated for the simultaneous removal and detoxification of phenanthrene (PHE) and benzo[a]anthracene (BAA) as well as heavy metals (HMs) (Cu, Zn, Pb and Ag).

The authors could include at the end of this paragraph, what type of samples were analyzed.

4-    Introduction

Line 65-66, pag 2

Include or mention specific examples with their respective bibliography, of the use of microorganisms in bioremediation.

What does he mean by the words but see?

5-    2.2.1. Phenanthrene and benz[a]anthracene biodegradation experiments

Flasks were sacrificed at regular intervals of 9 days

what does sacrifice mean?

6-    Line 207, page 5 (2.5.1. Phytotoxicity test)

without dilution and at three serial dilutions of 0.4 and 0.2 and 0.1),

Is it possible to be more specific?, and incorporate concentration units

7-    Lines 219-221, page 5 (2.5.2. Microtox® test)

Please include the concentration units of the samples tested.

8-    The tables and figures must be located, after the paragraphs where they are mentioned. Their presentation could also be improved regarding their quality, they are difficult to read.

9-    The intermediate metabolites of PHE and BAA after biodegradation by the strains detected by LC-MS are shown in Figure 2.

Include a new figure showing the HPLC of each sample, and identifying the major compounds, tentatively identified.

Mention or explain how you identified the major compounds, 1-phenanthrol (P1), phenanthrene 9,10-dihydrodiol (P2), 291 anthracene (B1), anthrone (B2), anthraquinone (B3), phthalic anhydride (B4) and phthalic 292 acid (B5)

After the suggested changes, the manuscript could be re-evaluated for acceptance.

Author Response

Dear Reviewer 2,

We appreciate your comments and corrections to improve the manuscript. Please find attached to this letter the files of the revised version of the abovementioned manuscript, to be considered for publication. We checked the whole manuscript again for typing errors to improve the manuscript. Changes in the manuscript have been marked in yellow.

Detailed comments to each of the reviewers’ remarks are given below. The original reviewers´ remarks are written in regular font while our answers/ statements are written in bold. The paper has been revised by proofreading services.com.

REVIEWER 2

Simultaneous heavy metal-polycyclic aromatic hydrocarbon removal by native Tunisian fungal species.

 Some suggestions are mentioned below

1-    Abstract

Please include the strain number of each of them

Strain accession numbers were included

 2-    2.1. Chemicals and strains

 Please include more information about the strains,

A paragraph with additional information about the strains has been added (l10-l14)

Define once what MH means

(MH667655.1) is the accession number of the strain in the GenBank, the two letters (MH) is a nucleotide prefixes assigned by the GenBank. It is a standard and conventional format [two-letter alphabetical prefix][six digits][.][version number].

3-    Lines 21-22, in pag1

 ……were investigated for the simultaneous removal and detoxification of phenanthrene (PHE) and benzo[a]anthracene (BAA) as well as heavy metals (HMs) (Cu, Zn, Pb and Ag).

 The authors could include at the end of this paragraph, what type of samples were analyzed.

 “ in kirk´s media was added”, also in accordance with the comment of the first reviewer.

4-    Introduction

 Line 65-66, pag 2

 Include or mention specific examples with their respective bibliography, of the use of microorganisms in bioremediation.

Specific examples were included (Lines 72-81)

What does he mean by the words but see?

 The words “but see” is a typing error it was deleted

5-    2.2.1. Phenanthrene and benz[a]anthracene biodegradation experiments

 Flasks were sacrificed at regular intervals of 9 days.  what does sacrifice mean?

 In this experiment we prepared 3 different flasks (independent replicates) for each interval time, 3 from the beginning flasks for time 0, 3 flasks for time 9 and 3 flasks for time 18. In addition 3 heat inactivated flasks were prepared at each time. “Flasks were sacrificed” means that all the content of the flask was used for analysis.

6-    Line 207, page 5 (2.5.1. Phytotoxicity test)

without dilution and at three serial dilutions of 0.4 and 0.2 and 0.1),

Is it possible to be more specific?, and incorporate concentration units

 7-    Lines 219-221, page 5 (2.5.2. Microtox® test)

Please include the concentration units of the samples tested.

The toxicity was expressed as EC50 (%) defined as the concentration which provokes a 50% light reduction on A. fischeri after 5 and 15 min of exposure. 

8-    The tables and figures must be located, after the paragraphs where they are mentioned. Their presentation could also be improved regarding their quality, they are difficult to read.

 The quality of the figures has been changed. The allocation at the end of the manuscript is a rule of JoF journal, but in the final version it supposed to be correctly placed, when mentioning in the text.

9-    The intermediate metabolites of PHE and BAA after biodegradation by the strains detected by LC-MS are shown in Figure 2.

 Include a new figure showing the HPLC of each sample, and identifying the major compounds, tentatively identified.

The identification was made by comparing the ionized molecular mass of the signals detected in the UHPLC-MS and the molecular mass of the metabolites of the previously reported metabolic pathways of degradation, in addition to considering the expected reaction mechanisms for the microbial degradation of these compounds by if there are unreported metabolic intermediates, the errors calculated for this comparison are reported in the supplementary material and not exceeding 20 ppm.

 Mention or explain how you identified the major compounds, 1-phenanthrol (P1), phenanthrene 9,10-dihydrodiol (P2), 291 anthracene (B1), anthrone (B2), anthraquinone (B3), phthalic anhydride (B4) and phthalic 292 acid (B5)

  Since the number of samples is high and different experimental variables were tested, it is not possible to design a single figure that represents all the chromatographic analyzes that were carried out, also considering that it is routine experimental work, we believe that the requested figure does not represent a significant contribution to this work

 After the suggested changes, the manuscript could be re-evaluated for acceptance.

Reviewer 3 Report

Dear сolleagues! This is very good job. Congratulations! However, I have a few questions/suggestions to improve the manuscript.

1. You have used different metal salts. Are you sure that the different effects of Cu and Zn, Pb or Ag resulted from the effects of cations, but not anions (SO4 and NO3, respectively)?

2. Half of the content of flasks was used for the analysis of enzymes, biomass, etc., the other half was used for the analysis of PAHs. Are you sure that PAHs and biomass are evenly distributed in both halves? PAHs are poorly soluble in water. In my experience, it is better to use the full volume of the flask for PAH analysis.

3. How was the sorption of PAHs by mycelium taken into account?

4. Figures need improvement. It's hard to see now.

Author Response

Dear Reviewer 3,

We appreciate your comments and corrections to improve the manuscript. Please find attached to this letter the files of the revised version of the abovementioned manuscript, to be considered for publication. We checked the whole manuscript again for typing errors to improve the manuscript. Changes in the manuscript have been marked in yellow.

Detailed comments to each of the reviewers’ remarks are given below. The original reviewers´ remarks are written in regular font while our answers/ statements are written in bold. The paper has been revised by proofreading services.com.

REVIEWER 3

Comments and Suggestions for Authors

Dear сolleagues! This is very good job. Congratulations! However, I have a few questions/suggestions to improve the manuscript.

  1. You have used different metal salts. Are you sure that the different effects of Cu and Zn, Pb or Ag resulted from the effects of cations, but not anions (SO4 and NO3, respectively)?

All the experiments were carried out in a Kirk’s media (rich on mineral salts) prepared with half-strength seawater which also contain considerable amount of salts (10.7 g/L of MgCl2, 5,4g of MgSO4.7H20, etc…). As well, the effect of heavy metals was tested in the same medium the only difference was the presence of metals, we used metal salts to improve metal solubility in liquid media but the amount of anions after dissociation in water will be negligible comparing with that already exist.

  1. Half of the content of flasks was used for the analysis of enzymes, biomass, etc., the other half was used for the analysis of PAHs. Are you sure that PAHs and biomass are evenly distributed in both halves? PAHs are poorly soluble in water. In my experience, it is better to use the full volume of the flask for PAH analysis.

The content of the flasks was well mixed and cut tips were used for collection. It would be better to use the full volume for PAH analysis, but the experiment carried out in one flask to minimize errors and to better understand the correlation between PAHs degradation and the other parameters.

  1. How was the sorption of PAHs by mycelium taken into account?

Incubation of media containing PAHs with inactivated mycelium was used to evaluate PAHs sorption.

  1. Figures need improvement. It's hard to see now.

Figures have been improved.

Round 2

Reviewer 2 Report

The authors have responded and considered the suggested changes, so I suggest the manuscript should be accepted for publication in its current state.